# Reproducibility Study of "Are Your Explanations Reliable?" Investigating the Stability of LIME in Explaining Text Classifiers by Marrying XAI and Adversarial Attack

## Reproducibility Summary

**Abstract**

This work investigates the reproducibility of "Are Your Explanations Reliable?" Investigating the Stability of LIME in Explaining Text Classifiers by Marrying XAI and Adversarial Attack by Burger et al. (2023). Our objective is to replicate and verify this paper's findings. The provided code by the authors is utilised as a foundation, missing segments and substantial additions are implemented by us. Our work suggests that the inherent instability claim is only partially reproducible due to unspecified hyperparameters in the paper. Nonetheless, we successfully reproduced and extended the results regarding the choice of RBO as similarity measure. Lastly, the third claim was partially reproducible due to constrained computational resources. However, we could verify the third claim by observing similar trends on a small subset of the test data. In conclusion, all claims are supported in varying degrees through our reproducibility study.

## 1 Introduction

In the rapidly evolving landscape of explainable AI (XAI), Local Interpretable Model-agnostic Explanations (LIME) (Ribeiro et al., 2016) has emerged as a widely utilised tool integrated into machine learning applications across diverse fields, ranging from healthcare (Barr Kumarakulasinghe et al., 2020) to finance (Tan et al., 2023). In these high-stake fields, where the interpretability of AI models is crucial, ensuring the stability of LIME is essential for its reliability.

In the paper: "Are Your Explanations Reliable?" Investigating the Stability of LIME in Explaining Text Classifiers by Marrying XAI and Adversarial Attack by Burger et al. (2023), the stability of LIME in the context of text data is researched. Furthermore, the novel algorithm XAIFooler is proposed. XAIFooler perturbs text inputs and manipulates explanations, probing the stability of LIME in handling textual changes. XAIFooler adheres to constraints to preserve text semantics and original predictions with small perturbations. It utilises Rank-biased Overlap (RBO) (Webber et al., 2010) as the similarity measure between explanations in its optimisation.

Our objective is to replicate the findings of Burger et al. (2023), thereby validating their claims. Specifically, our contributions include:

- Reproducing the original experiments and discussing the degree of similarity between the original and reproduced results.

- Streamlining the codebase for improved clarity and functionality by addressing initial non-functioning and non-existing code, implementing necessary modifications and enhancing overall structure with the addition of comments and documentation.

- Extending the investigation through additional research to further substantiate the claims.

## 2 Scope of Reproducibility

This work investigates the reproducibility of the paper by Burger et al. (2023), which addresses the instability of LIME. Namely, they state that previous attempts at making black-box models transparent in their decision-making, employ a baseline by Sinha et al. (2021) that overestimates this instability. Furthermore, they highlight the challenges posed by text-specific constraints and introduce a novel algorithm, XAIFooler, to perturb text inputs and manipulate LIME's explanations. Previous research in modifying these explanations utilised $L_2$ and Center of Mass (COM) (Sinha et al., 2021) as a similarity measure between the perturbed and original explanation, whereas XAIFooler employs RBO.

We aim to reproduce the results and verify the main claims of the paper, which are as follows:

1. LIME exhibits inherent instability in explaining text classifiers.

2. RBO is the most suitable similarity measure for text perturbation optimisation compared to Jaccard Similarity, Spearman's $\rho$, $L_2$ and COM.

3. XAIFooler significantly outperforms all baselines, including inherent instability, random, $L_p$ and Location of Mass (LOM), by large margins in its ability to manipulate LIME's explanations with high semantic preservability.

Additionally, we improve and extend the original code to ensure that all experiments can be executed through one main Python file per experiment. The code is made interpretable by providing clarifications via comments, thereby enhancing its accessibility and transparency. Secondly, we provide an extension on the experiment corresponding to claim 2. This addition aims to generalise the claim to a higher extent.

## 3 Methodology

Our methodology closely follows the original paper's approach, utilising the provided code in their GitHub repository[1] as a foundation. Due to initial functionality issues, we make necessary adjustments to ensure the code's proper functionality while maintaining alignment with the original described procedures.

### 3.1 Model Descriptions

In compliance with the original study, LIME is utilised to explain black-box models. It employs an explainable surrogate model: Logistic Regression with Elastic Net Regularisation, trained using Stochastic Gradient Descent. LIME is applied to three (black-box) models featured in the initial experiments: DistilBERT (Sanh et al., 2019), BERT (Devlin et al., 2019) and RoBERTa (Liu et al., 2019). These models[2] are trained by Burger et al. (2023) on 80% of the datasets. We deploy these pretrained models, with the remaining 20% constituting the test set.

Finally, XAIFooler is implemented as proposed in the original study, functioning as an adversarial attacker on LIME. This greedy-based algorithm manipulates text instances to explore LIME's robustness beyond its intrinsic instability by leveraging word importance as determined by the surrogate model.

### 3.2 Datasets

In accordance with the original study, our experiment involves three datasets: Internet Movie Database (IMDB) (Maas et al., 2011), Symptom to Diagnosis (S2D) (from Kaggle) and Gender Bias (GB) (Dinan et al., 2020). The dataset specifications as utilised in our experiments can be found in Table 1.

---

[1] Repository of original paper: `https://github.com/cburgerolemiss/xaifooler`
[2] Models available at: `https://huggingface.co/thaile`

Table 1: Dataset specifications for the reproduced experiments.

| Dataset | Test Size | Batch Size[*] | # of Tokens | # of Labels |
|---|---|---|---|---|
| Movie reviews (IMDB) | 4 | 64 | 230 | 2 |
| Symptom to Diagnosis (S2D) | 40 | 128 | 29 | 21 |
| Gender Bias (GB) | 200 | 256 | 11 | 2 |

[*] During LIME's surrogate model training

### 3.3 Hyperparameters

The hyperparameters are selected to be in correspondence with the original paper. Whenever values cannot be found in the paper, we follow the hyperparameters provided by the code. If these are unspecified in the code as well, we apply the default arguments as per the official documentation of the package. Furthermore, if the code contradicts the paper, we prioritise the values in the paper.

Certain hyperparameters require additional clarification. One such value is $k$, representing the top number of words considered crucial in a generated explanation. These words are protected from perturbation by XAIFooler, and they must rank differently for a perturbed document's explanation to unveil instability. The values of $k$ chosen for IMDB, S2D and GB are 5, 3 and 2, respectively. We suspect these values are relative to the average document length in each dataset, implying that longer documents might contain more important features for classification.

For claim 2, the original paper conducted an experiment to determine the hyperparameter $p$ in RBO by varying the fraction of combined mass to obtain the corresponding value for $p$. For each dataset, the corresponding amount of top $k$ features is utilised. We implemented this experiment and the results are shown in Figure 1. Our reproduction also illustrates that for 90% (IMDB), 95% (S2D), 95% (GB) mass concentrating in the top $k$ features corresponds with $p$ values of 0.75, 0.49, and 0.32, respectively. This aligns exactly with the original hyperparameter investigation.

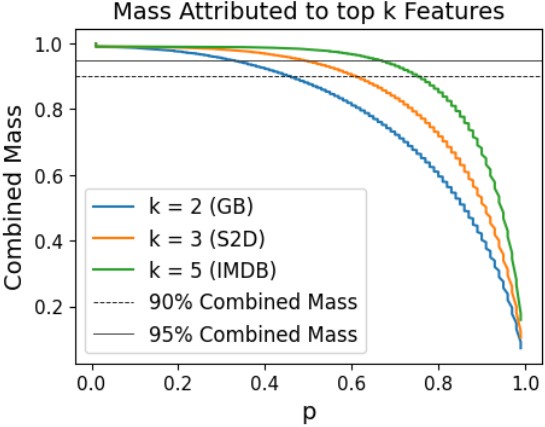

Figure 1: Selecting the hyperparameter $p$ for RBO by adjusting the fraction of combined mass attributed to the top $k$ features across dataset specific values of $k$. Horizontal lines are added to highlight the points of intersection with 90% and 95% combined mass.

### 3.4 Experimental Setup and Code

In our exploration of the three distinct claims, we have encountered diverse challenges. First of all, an environment must be established to resolve some non-functioning modules and issues related to bugs need to be addressed. For claim 1 and claim 2 the required functions are only partially present in the authors' provided code, hence substantial code rewriting is required. Regarding claim 3, most of the code is provided,

except for the calculation of one metric (PPL). The modified code for our experiments can be found at `https://anonymous.4open.science/r/reproml24`.

### 3.4.1 Inherent Instability

Claim 1 is examined in similar fashion to the original study. Explanations are generated for documents using various sampling rates ($n$). Thereafter, each explanation is compared with the reference explanation obtained using the default sampling rate ($n$=5000). The sampling rate determines the number of samples generated by LIME. Each sample consists of the original document with $k$ words randomly removed. This evaluation is conducted on all target models in combination with each dataset.

The original paper does not explicitly state the hyperparameters for the inherent instability experiment. Therefore, to maintain consistency, we assume that most values are identical to the experiment for the third claim. This includes Logistic Regression as surrogate model for LIME. Moreover, the range for the sampling rate is described as 1,000-7,000 in the paper. However, the step size (250) is not explicitly mentioned in the paper and needs to be deduced from their figure containing the results, resulting in 25 distinct sampling rates. The minimum and maximum values for $k$ are set to the default, meaning that at least one and at most all words in a document are removed to create a sample. To save computational resources, the number of documents per sampling rate is set to 50, 25 and 5 for GB, S2D and IMDB, respectively. This amount of documents leads to an average computing time of one hour, resulting in a total of nine hours for the complete inherent instability experiments. Due to the consumed resources for the remaining claims, the decision was therefore made to not employ more documents than the aforementioned values. These documents are assumed to be combined as an average. Lastly, the similarity measure is assumed to be RBO since it takes feature order into account (the explicit ranking of features can lead to different explanations, although the features may be the same) and aligns most with the findings in the original study, see Section 3.3.

### 3.4.2 Comparison of Similarity Measures

In the original paper, the following similarity measures are compared: RBO, Jaccard Index (Murphy, 1996) and Spearman's $\rho$ (Spearman, 1904). Additionally, alternative metrics, namely $L_2$ and Center of Mass (COM) (Ghorbani et al., 2019), are only compared theoretically. Therefore, to align with the original paper, we discard these metrics as well.

To validate their findings, we undertake the task of implementing the experiment, as the code was omitted from the provided resources. We use an arbitrary list of 50 distinct features and subjected this list to 50,000 random shuffles with seed 1212. For every permutation, the similarity between the current shuffled list and the original list is calculated for the similarity measures. Specifically, for the latter two, the similarity calculation considered only the top $k$ features. In contrast, the RBO similarity is computed across the entire list, with its hyperparameter $p$ specifying how much influence the top $k$ features have.

### 3.4.3 Comparison of XAIFooler Against Baselines

In order to showcase XAIFooler's effectiveness in manipulating LIME's explanations on DistilBERT, BERT and RoBERTa, its performance is compared against baselines established in the original paper. For further details, refer to Appendix A.1. While most of the code was provided by the authors, it initially faced functionality issues. Through bug fixing, code structuring and addition of missing code for the PPL metric, we successfully made the files executable.

The PPL metric is defined in the same way as described by the original paper. Namely, it consists of the negative log likelihood returned by GPT-2 for a given input sequence. As this evaluation metric was not present in the original code, a separate function was created that is called whenever evaluation occurs. This function is adapted from Huggingface's documentation on perplexity[3].

---

[3]`https://huggingface.co/docs/transformers/perplexity`

It is worth noting that the experiment utilises drastically fewer test set samples compared to the original paper due to demanding computational costs. The specific sizes of the test set can be found in Table 1. As for the evaluation of the different methods, we employ the metrics as described by Burger et al. (2023).

### 3.4.4 Additional Comparisons with RBO

As an additional experiment, other similarity measures are investigated to support claim 2, one of which is Cosine Rank Similarity (CosRank) (Dinu & Ionescu, 2012). This measure is designed to compare rankings, giving more importance to higher-ranked features. None of the measures that RBO is compared with in the original paper fulfil more than two of the four desired properties. The original paper provides these properties for comparing two ranked explanation lists. CosRank does satisfy all four properties, as can be seen in Table 2. Therefore, we compare RBO to CosRank to investigate whether RBO can be outperformed by a different measure that satisfies all properties.

Additionally, the original paper mentions both Spearman's $\rho$ and Kendall's $\tau$ (Kendall, 1938) as a measure of Rank Correlation (Galton, 1889), but only Spearman's $\rho$, tested solely on the top $k$, is utilised in their experiment. The step-wise behaviour of the returned similarities is considered disadvantageous, as it leads to poorer delineation between the important features. For this reason, we test Spearman's $\rho$ and Kendall's $\tau$ on the full list.

Table 2: Theoretical inspection of CosRank based on the desired properties stated by the original paper.

| Feature / Measure | Positional Importance | Feature Weighting | Disjoint Features | Unequal-Length List |
|---|---|---|---|---|
| Jaccard Index | | ✓* | ✓ | ✓ |
| Kendall's $\tau$ | ✓ | ✓* | | |
| Spearman's $\rho$ | ✓ | ✓* | | |
| Lp | | ✓ | ✓ | |
| Center of Mass | | ✓ | ✓ | |
| RBO | ✓ | ✓ | ✓ | ✓ |
| CosRank | ✓ | ✓* | ✓ | ✓ |

\* Customised formula needed

### 3.5 Computational Requirements

The average individual runtime per experiment can be found in Table 3. In total, approximately 550 hours of GPU are utilised. The GPU consists of a cluster with an NVIDIA A100 GPU run on SURF's Snellius.

Table 3: Experiment runtimes and resource utilisation

| Experiment | Hardware | Average Runtime | # of Experiments |
|---|---|---|---|
| Claim 1 | GPU | 1 hour | 9 |
| Claim 2 | CPU | 30 seconds | 1 |
| Claim 3 | GPU | 12 hours | 45 |

## 4 Results

The reproduced experiments display minimal differences with the original study considering claim 2 and 3. Although the results for claim 1 differ more significantly, they do not display outcomes that may refute the original claim. The additional experiment further substantiates claim 2 by displaying that RBO consistently outperforms all other measures. Furthermore, a short qualitative evaluation is provided in Appendix B.

### 4.1 Inherent Instability

Figure 2 displays the results for the experiment that examines claim 1. The lower the intersection proportion (RBO score), the greater the instability. In general, the document size appears to be the most influential for the instability of LIME, whereby IMDB is most unstable. Moreover, as the document size increases, the difference between varying sampling rates within one model increases as well. This can be seen in the figure, as IMDB, which has the biggest average document size (Table 1), shows the biggest change in the intersection proportion. And GB, which has the smallest average number of tokens per document (Table 1), shows little change in the intersection proportion. These trends do not directly align with the original paper's results where the intersection proportion appears more stable over the varying document sizes. Nonetheless, claim 1 is supported, since our results indicate that LIME is inherently unstable.

The most promising sampling rates depend on the balance between the stability of LIME and the computational requirements. Therefore, the optimal choices are the lowest sampling rates at which the intersection proportions are stable. For GB, we estimate this value at approximately 2,000 since the intersection proportion does not fluctuate significantly starting at around 2,000. For the S2D dataset, DistilBERT's intersection proportion barely fluctuates from the beginning to the end. Thus, an optimal approach for this instance is to take the other model's preferences into account. For BERT and RoBERTa, a preferable sampling rate would be around 2,500. Hence, an overarching sampling rate of 2,500 might be appropriate for S2D. Lastly, IMDB exhibits the highest instability over the sampling rates. Although the stability appears to increase around a sampling rate of 5,000, the intersection proportion seems to fluctuate again around 7,000. Overall, the sampling rates align with the original paper, whereby the values for GB and IMDB are slightly higher with a difference of 500.

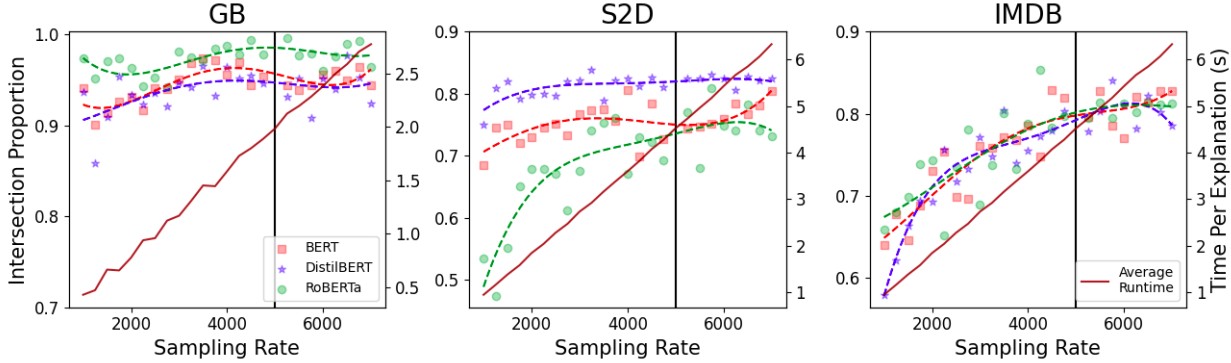

Figure 2: Inherent instability explanations by comparing the intersection proportion of explanations over various sampling rates to the default explanation with sampling rate 5,000.

### 4.2 Comparison Similarity Measures

To support claim 2, the authors of the original paper compared RBO with other metrics, namely Jaccard Index and Rank Correlation. This experiment led the authors to assert that RBO is the most suitable for this application. We agree, as other methods exhibit poorer discrimination between important features, and because the other methods results in zero similarities for the majority of permutations (Burger et al., 2023). In contrast, RBO has the ability to give weights to features outside the top k, leading to non-zero similarities and smoother assignment of similarities to permutations of features, as illustrated in Figure 3.

Figure 3 displays the comparison of four similarity measures. It can be seen that Spearman's $\rho$ and the Jaccard Index remain zero for over half the permutations. Thereafter, Spearman's $\rho$ climbs up rapidly in a step-wise manner and has perfect similarity for the final permutation. The Jaccard Index shows a similar step-wise increase, but with fewer steps and a maximum similarity lower than 0.7. In contrast, the RBO measures display a more smooth increase in the similarity and the similarity is non-zero from the beginning.

The maximum value for both RBO measures is approximately 0.8. On average, the RBO measure with 80% mass on the top 5 is smoother than the RBO with 90% mass, albeit slightly higher.

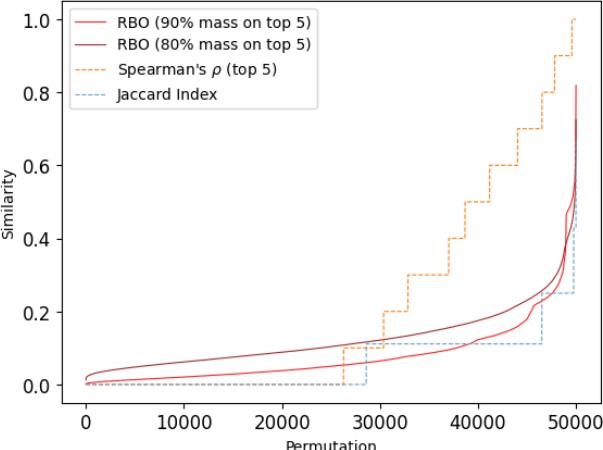

Figure 3: The comparison of the similarity measures for 50,000 permutations, with the similarity values ordered from low to high.

## 4.3 Comparison XAIFooler Against Baselines

In order to verify claim 3, the black-box models are subjected to perturbations. This involves applying XAIFooler as well as the four baselines individually across each dataset. The outcomes of these experiments are presented in Table 4.

Table 4: Results of XAIFooler against other baselines. **Bold** and *cursive* values denote the best and second-best result (excluding "Inherency") per dataset and model, respectively. Arrow ↑ and ↓ denote the higher, the better and the lower, the better, respectively.

| Dataset/Method | DistilBERT | | | | | BERT | | | | | RoBERTa | | | | |
|---|---|---|---|---|---|---|---|---|---|---|---|---|---|---|---|
| | ABS↑ | RC↑ | INS↓ | SIM↑ | PPL↓ | ABS↑ | RC↑ | INS↓ | SIM↑ | PPL↓ | ABS↑ | RC↑ | INS↓ | SIM↑ | PPL↓ |
| **IMDB** Inherency | 5.50 | 0.78 | 0.90 | 1.00 | 33.20 | 18.00 | 0.38 | 0.80 | 1.00 | 33.20 | 13.75 | 0.65 | 0.70 | 1.00 | 33.20 |
| Random | 10.75 | *0.83* | 0.75 | 0.87 | 112.14 | 21.75 | 0.45 | 0.70 | 0.86 | 133.08 | 9.75 | *0.80* | *0.75* | 0.86 | 119.51 |
| LOM | 2.50 | 0.25 | 0.85 | **0.97** | 36.24 | 11.50 | 0.53 | 0.75 | **0.96** | **39.90** | 11.50 | **0.93** | *0.75* | **0.97** | 36.37 |
| $L_p$ | *20.00* | 0.55 | **0.65** | 0.89 | 89.37 | *31.67* | **1.00** | *0.67* | 0.89 | 91.07 | *35.25* | 0.70 | *0.75* | 0.87 | 114.93 |
| **XAIFooler** | **26.00** | **0.85** | *0.70* | *0.92* | *40.63* | **34.50** | *0.65* | **0.40** | *0.93* | *41.06* | **50.75** | 0.63 | **0.60** | *0.92* | *49.97* |
| **S2D** Inherency | 1.13 | 0.21 | 0.92 | 1.00 | 11.09 | 1.50 | 0.29 | 0.89 | 1.00 | 11.09 | 2.38 | 0.20 | 0.78 | 1.00 | 11.09 |
| Random | 1.55 | 0.31 | 0.88 | 0.84 | 76.16 | 2.24 | 0.30 | 0.83 | 0.84 | 85.98 | **3.41** | **0.41** | **0.77** | 0.84 | 75.63 |
| LOM | 0.89 | 0.18 | *0.87* | **0.93** | 19.53 | 0.97 | 0.20 | 0.89 | **0.94** | 17.64 | 1.94 | *0.39* | 0.82 | **0.93** | 22.14 |
| $L_p$ | *1.58* | *0.33* | 0.90 | 0.83 | 67.12 | **2.85** | *0.41* | *0.81* | 0.83 | 78.68 | *3.33* | **0.41** | 0.80 | 0.84 | 62.62 |
| **XAIFooler** | **2.17** | **0.45** | **0.78** | *0.88* | *44.00* | *2.64* | **0.46** | **0.73** | *0.89* | *32.31* | 2.38 | 0.35 | *0.78* | *0.90* | *26.71* |
| **GB** Inherency | 0.23 | 0.07 | 0.94 | 1.00 | 163.12 | 0.39 | 0.17 | 0.92 | 1.00 | 164.63 | 0.31 | 0.14 | 0.94 | 1.00 | 165.70 |
| Random | *1.05* | *0.27* | *0.75* | 0.81 | 662.93 | *0.98* | 0.21 | *0.77* | 0.81 | 670.50 | *0.72* | *0.21* | 0.80 | 0.81 | 628.68 |
| LOM | 0.60 | 0.22 | *0.75* | **0.87** | 329.98 | 0.68 | 0.23 | 0.82 | **0.87** | 350.12 | 0.45 | 0.16 | *0.76* | **0.87** | 374.82 |
| $L_p$ | 0.64 | 0.22 | 0.92 | 0.80 | 632.72 | 0.66 | *0.24* | 0.92 | 0.80 | 627.17 | 0.67 | *0.21* | 0.90 | 0.81 | 631.52 |
| **XAIFooler** | **1.59** | **0.50** | **0.65** | *0.84* | *408.83* | **1.71** | **0.53** | **0.65** | *0.84* | *416.34* | **1.56** | **0.53** | **0.65** | *0.84* | *436.29* |

Notably, the inherency baseline, as mentioned in the original paper, is excluded from the comparison of perturbation methods. We assume that this stems from the fact that some metrics would convey incorrect information. For instance, similarity will always equal 1.0 when the document is not adjusted and perplexity

would assess a human's naturalness in contrast to the model's perturbation capabilities. Consequently, we have omitted the inherency baseline from our analysis as well.

Moreover, as can be seen in Table 4, XAIFooler consistently outperforms the other perturbation methods for almost all combinations of models and datasets concerning ABS, RC and INS. In terms of SIM and PPL, it persistently secures the second position. Although LOM exhibits slightly better performance than XAIFooler on SIM and PPL, it significantly lags behind on the remaining three metrics.

### 4.4 Additional comparisons with RBO

To further support claim 2, we compare RBO with CosRank, Kendall's $\tau$ and Spearman's $\rho$ on both the top 5 and the full list.

Figure 4 illustrates that the other similarity measures behave differently from RBO. Kendall's $\tau$ returns, similarly to Spearman's $\rho$, zero similarities for over half of the permutations. Furthermore, the highest similarity of Kendall's $\tau$ is only 0.4, against RBO's 0.8. Finally, although CosRank's shape resembles the increase of similarity calculated by RBO with 80% mass on top 5, its returned similarities are at least 0.4 for most of the permutations. As a result, this measure gives a high similarity for rankings which are significantly different in order. This leads to a smaller range to distinguish between the similarities of the permutations.

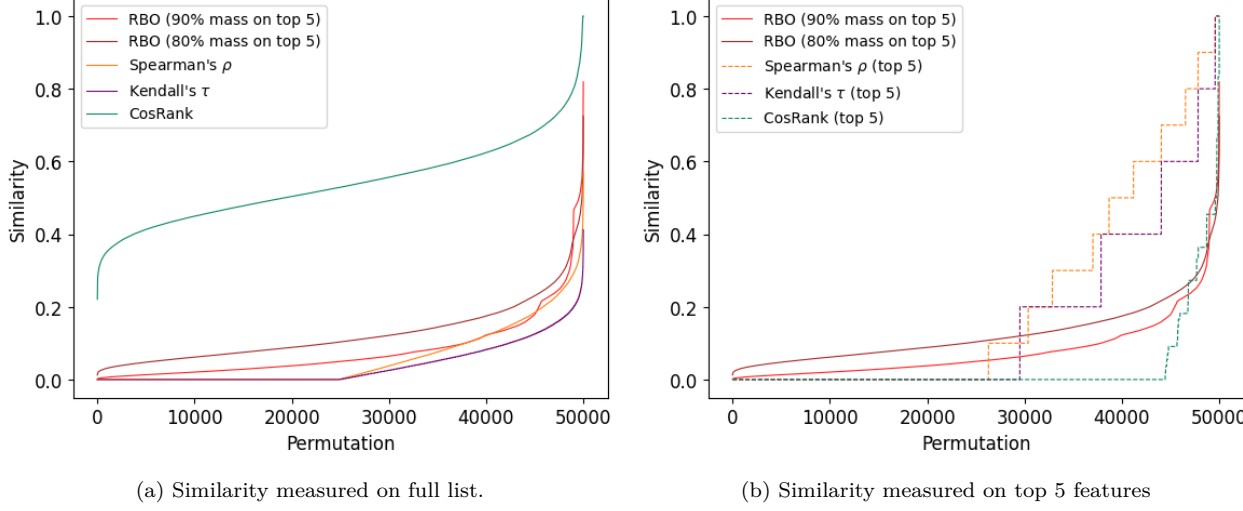

(a) Similarity measured on full list.

(b) Similarity measured on top 5 features

Figure 4: The comparison of the similarity measures for 50,000 permutations, with the similarity values ordered from low to high.

## 5 Discussion

Throughout this work, three experiments are conducted to support the main claims as explored by Burger et al. (2023). Our reproduced results support most of their claims, even when utilising a reduced number of documents. Specifically, our findings for claim 2 are nearly identical to the results reported in the original paper. Consequently, the reasons behind the authors' selection of RBO as similarity measure for XAIFooler's optimisation, along with the corresponding $p$ values become evident in Figure 3 as well. Furthermore, similarity measures from the additional experiment return values that diverge significantly from those obtained through RBO. Specifically, Spearman's $\rho$ and Kendall's $\tau$ yield zero similarity for most permutations, while CosRank produces higher values. The latter results in a narrower range of similarities, limiting their ability to effectively convey distinctions between rankings. Consequently, this displays that RBO is more suitable as a similarity measure and proves claim 2.

The comparison of XAIFooler against the baselines greatly corresponds with the findings in the paper. The absolute values closely resemble the original paper's values and more importantly, they display similar

trends. Specifically, XAIFooler is the best or second best performing method across all the selected datasets and models. Moreover, despite the fact that LOM obtains the highest results on semantic preservability, XAIFooler always achieves the second highest results. Thus, we conclude that XAIFooler outperforms all baselines based on all considered metrics with high semantic preservability, as stated in claim 3. This is in correspondence with the original paper.

We experienced severe complications reproducing the inherent instability experiment. The experiment suffers from a remarkable lack of known hyperparameters. For instance, the original paper mentions that an 'arbitrary' number of documents is utilised, and it is found that numerous outliers result in significantly different scores depending on the employed quantity. Additionally, the number of tokens that are removed to generate samples within LIME ($k$) is unclarified in the paper. We opted for the default, implying that $k$ ranges from one to all tokens being removed. This introduces a variable that can notably impact the explanations. Establishing a stable value for $k$ could therefore potentially mitigate these variations. Furthermore, the similarity measure is unspecified. This impacts the outcomes significantly, since another similarity measure might lead to extremely different similarity scores. Lastly, additional hyperparameters that are less influential on the final results, were missing. Therefore, a search algorithm that accounts for all possible combinations of hyperparameters would be infeasible, leading us to rely on conjectures. The lack of specified hyperparameters leads to a crucial difference in the absolute results compared to the original study. Moreover, the sampling rates for claim 3 are deduced from these results in the original paper. Therefore, the sampling rate utilised in claim 3 does not correspond with our findings about the inherent instability of LIME. Nonetheless, claim 1 can still be supported by our results since we find LIME to be inherently unstable.

Finally, we want to address the environmental impact of the experiments. Despite reducing the size of the test set, our replication of the paper necessitates approximately 550 GPU hours. Consequently, the amount of computing resources may lead to an increasing amount of carbon emissions depending upon the provider's specifications, see Appendix C for an example. In our case, the provider (SURF's Snellius) operates on 100% green energy and therefore the carbon emissions and overall environmental pollution are negligible (van der Tak et al., 2021). Nonetheless, one should be mindful about the amount of computing resources, particularly as our research indicates that comparable results can be attained with a reduced sample size, thus requiring fewer computational resources.

In conclusion, all claims are supported through our reproducibility study. Nevertheless, we would argue that the paper's reproducibility is limited, considering that the experiment to support claim 1 suffers from a significant amount of unspecified hyperparameters in the original paper. Moreover, while the other two claims, including the main claim, exhibit a notable level of reproducibility, achieving this requires substantial understanding, effort and computational investment.

### 5.1   What Was Easy

The original paper is easily understandable and mentions most of the values for the hyperparameters for the main experiment. Furthermore, it has appendices about the reproducibility details and comprehensive explanations of the similarity measures. Additionally, a substantial portion of the code is provided.

### 5.2   What Was Difficult

The process of getting the code to function was time-consuming due to numerous bugs, lack of clarity, absence of an established environment, and missing code segments. Additionally, the paper lacked certain hyperparameter details or exhibited inconsistencies with the code specifications, necessitating educated guesses and leading to challenges in result reproduction. Moreover, substantial computational resources were required, as we frequently encountered out-of-memory errors and the total GPU hours spent for all experiments reached approximately 550 hours, excluding the time invested in addressing code-related issues. Lastly, we noticed that the Git repository got updated while we already started working on reproducing the paper. These updates mainly consisted of fixing bugs. However, not all bugs were taken care of, leading to code that was still non-functional. Besides, certain bugfixes did not fix the bug, which increased the difficulty of the reproduction process.

## 5.3 Communication with Original Authors

In an effort to conduct a fair and comprehensive assessment of the original research, we initiated communication with the three authors via the email addresses provided in the original paper. Our email was crafted to inquire about specific aspects of the paper and its associated code. Unfortunately, one of the provided email addresses rejected our communication attempt and from the other two authors we did not receive responses despite our efforts.

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

# A    Implementation details

## A.1    XAIFooler

XAIFooler is compared to the baselines established in the original paper. The first baseline consists of the inherent instability in LIME, as described in Section 3.4.1. Another baseline is utilised in order to display the effectiveness of XAIFooler's greedy-search approach in determining perturbations. Thus, instead of the greedy-search, this second baseline involves the random selection of words for perturbation and the random designation of a replacement candidate from a list of nearest neighbours.

Furthermore, the final baselines include different perturbation methods, such as Location of Mass (LOM) and $L_p$ (Sinha et al., 2021). While these methods share similarities with XAIFooler, they deviate in their choice of similarity function for comparing explanations. Specifically, the former employs COM, while the latter implements $L_2$ instead of RBO.

The utilised evaluation metrics applied for the comparison between XAIFooler and the baselines consist of: absolute change in ranking orders (ABS↑), rank correlation (RC↑), intersection ratio (INS↓), semantic similarity (SIM↑) and perplexity score (PPL↓). Arrow ↑ and ↓ denote the higher, the better and the lower, the better, respectively.

# B Qualitative results

This section provides a qualitative analysis of the influence of XAIFooler on LIME's explanations. We compare these explanations for the original documents to the explanations for the perturbed version of the document. We provide two examples per dataset in which XAIFooler's performance becomes clear. Particularly, we investigate how the features in the top $k$ shift when the document is perturbed, whereby the top $k$ features are not allowed to be perturbed by XAIFooler. Similarly to the main experiment, $k$ is 5, 3 and 2 for IMDB, S2D and GB, respectively.

## B.1 IMDB

Figure 5a displays the comparison of a movie review to the same review with sixteen perturbations. The explanation for the original review shows that 'wonderful', 'enjoyed', 'love', 'years' and 'sadly' make up the top 5 features in LIME's explanation. Although the top 2 remains the same for the perturbed document, 'love' has been demoted from the top 3 to the top 4 and 'years' and 'sadly' are not present in the top 5 anymore. This demonstrates that XAIFooler can successfully alter the explanation of LIME through perturbations. Moreover, the semantic meaning does not appear to have been altered significantly.

Figure 5b displays the results for another movie review with fewer perturbations: only three. Although the top 5 features does remain the same the internal order has shifted as follows: $1 \rightarrow 3$, $2 \rightarrow 4$, $3 \rightarrow 2$, $4 \rightarrow 1$, $5 \rightarrow 5$. Solely 'suspenseful' remains at the same spot in the top 5. Therefore, XAIFooler has successfully perturbed the explanations, but with a slightly lower impact than for the review in Figure 5a. This presumably stems from the fact that the document has been altered less heavily.

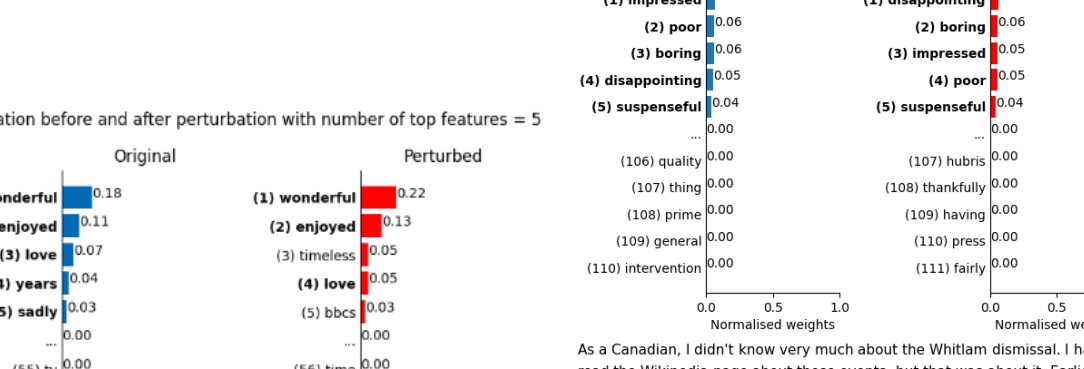

Explanation before and after perturbation with number of top features = 5

Original                Perturbed

(1) wonderful  0.18     (1) wonderful  0.22
(2) enjoyed  0.11       (2) enjoyed  0.13
(3) love  0.07          (3) timeless  0.05
(4) years  0.04         (4) love  0.05
(5) sadly  0.03         (5) bbcs  0.03
...  0.00               ...  0.00
(55) tv  0.00           (56) time  0.00
(56) content  0.00      (57) fell  0.00
(57) br  0.00           (58) video  0.00
(58) tried  0.00        (59) eyre  0.00
(59) anticipation  0.00 (60) tv  0.00

0.0    0.5    1.0       0.0    0.5    1.0
Normalised weights      Normalised weights

~~Just~~ Exclusively ~~reading~~ lire the reviews of this wonderful BBC mini ~~series~~ serial reminded me how much I enjoyed this when it was first broadcast many years ago. At the time I ~~remember~~ remembers waiting with anticipation for the next installment and fell in love with ~~John~~ Joon Bowe and Janet McTeer ,two ~~talented~~ skilled ~~actors~~ actresses that we don't see enough of on TV or ~~cinema~~ cine these days. 

I have tried without ~~success~~ accomplishments to obtain several of the BBCs ~~fantastic~~ fabulous dramas from the past, including the 1973 version of Jane Eyre and the 1972 version of Anne of ~~Green~~ Greenery Gables, all wonderful timeless classic ~~stories,~~ tales, which sadly the BBC ~~seem~~ transpires to ~~have~~ has no ~~intention~~ intentions of releasing on video or DVD. If anyone learns otherwise I would love to hear from them.

In the meantime we will have to content ourselves with our ~~recollections~~ reminiscences of how wonderful they were.

(a) Document with 16 perturbations.

Explanation before and after perturbation with number of top features = 5

Original                    Perturbed

(1) impressed  0.07         (1) disappointing  0.06
(2) poor  0.06              (2) boring  0.06
(3) boring  0.06            (3) impressed  0.05
(4) disappointing  0.05     (4) poor  0.05
(5) suspenseful  0.04       (5) suspenseful  0.04
...  0.00                   ...  0.00
(106) quality  0.00         (107) hubris  0.00
(107) thing  0.00           (108) thankfully  0.00
(108) prime  0.00           (109) having  0.00
(109) general  0.00         (110) press  0.00
(110) intervention  0.00    (111) fairly  0.00

0.0    0.5    1.0           0.0    0.5    1.0
Normalised weights          Normalised weights

As a Canadian, I didn't know very much about the Whitlam dismissal. I had read the Wikipedia page about those events, but that was about it. Earlier this year, when Canada went through a potential constitutional crisis (it fizzled out, thankfully) that might have led to intervention by our Governor-General, the Whitlam dismissal was mentioned in the press. In an effort to learn more, I ordered the DVD of this mini-series through EBay.

I was greatly impressed by how interesting the account was. As dramatic as events were, this could have been a very boring political drama. However, it was a pretty suspenseful mini-series. I was also impressed by how understandable it was, despite my lack of familiarity with Australian politics. It didn't take long to figure out who everyone was, and what their roles were.

Having said that, it is not an entirely impartial account. Malcolm Fraser is certainly portrayed as a rather ~~Machiavellian~~ Manipulative figure, who lets no person or thing get in the way of his quest to be Prime Minister. Gough Whitlam is portrayed in a more noble, almost saintly, light. However, the actor portraying Whitlam channels the nobility in such a way that it comes across more as ~~pomposity.~~ hubris. I thought that Sir John Kerr was portrayed in a fairly sympathetic manner.

I must warn people that the DVD is of very poor quality. I understand that it was made for television in the early 80s, but it would appear that no effort was made to restore the picture quality or sound quality. It was ~~very~~ enormously disappointing that no extras were added either. A documentary, or even some interviews with the historical figures, would have enhanced the experience, but there is nothing.

I highly recommend this mini-series for anyone interested in the real-life events.

(b) Document with 3 perturbations.

Figure 5: Comparison of top 5 features in LIME explanation for two original and perturbed documents from the IMDB dataset.

## B.2  S2D

Figure 6 displays the influence of XAIFooler on two examples from the S2D dataset. The changes in the sentences appear to preserve the semantic meaning. Figure 6a is an example of a document for which XAIFooler did not manage to alter the top features in LIME's explanation. This might be attributed to the fact that no more than one token was altered in the document. Moreover, we can see that features 4-8 changed positions.

Figure 6b displays the effect of two additional changes to the original document for this dataset. With three perturbations, XAIFooler does succeed in altering the top 3 features in LIME's explanation. Specifically, the features in the top 2 switched positions and the 'shakes' demoted from the third to fifth place in the ranking.

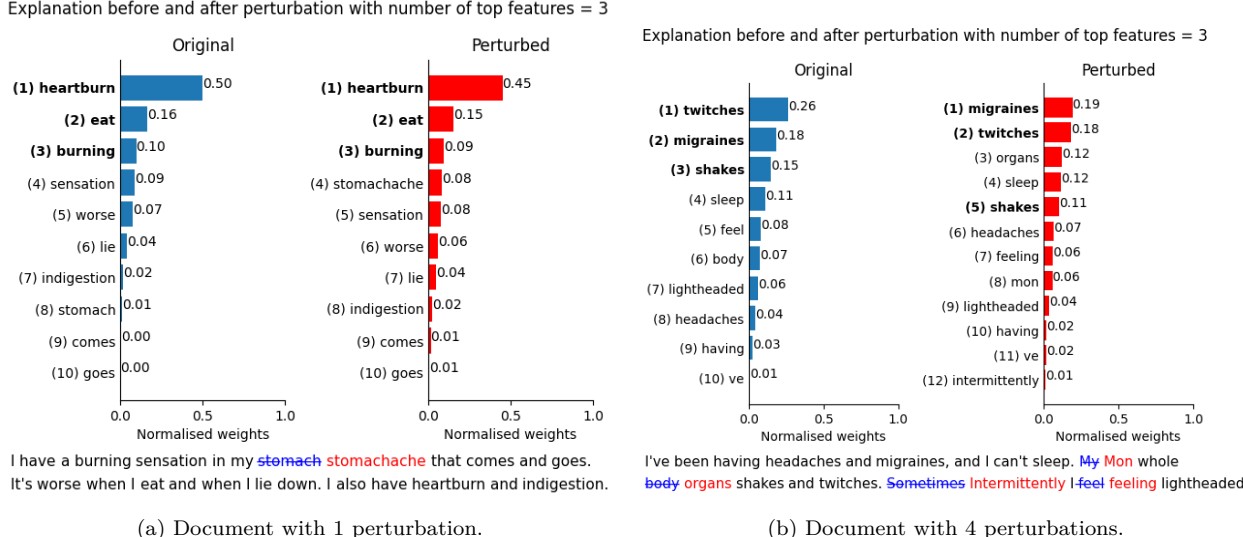

(a) Document with 1 perturbation.

(b) Document with 4 perturbations.

Figure 6: Comparison of top 3 features in LIME explanation for two original and perturbed documents from the S2D dataset.

## B.3  GB

Figure 7 demonstrates that the explanations of LIME are least altered by XAIFooler when the sentence is relatively short. The document in Figure 7a consists of 10 tokens, and two are altered by XAIFooler. Although this is 20% of the full document, the most important feature for the explanation remains the same. However, the second most important feature is demoted to the fifth place.

In Figure 7b it becomes more evident that XAIFooler is less impactful on smaller documents. This document is slightly longer than the sentence from Figure 7a, but the ranking has barely been altered. The only difference is that the second most important feature moves to the third place.

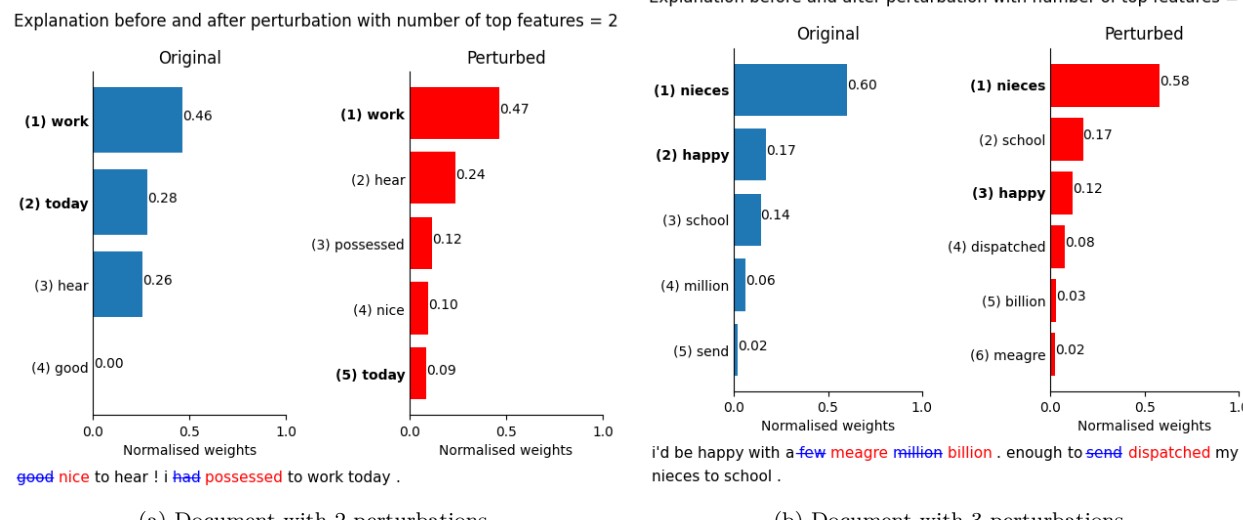

(a) Document with 2 perturbations.  (b) Document with 3 perturbations.

Figure 7: Comparison of top 2 features in LIME explanation for two original and perturbed documents from the GB dataset.

## C  Environmental impact

During the execution of the main experiment, we suffer from a significant number of GPU hours. Depending on the exact hardware, provider, and location of the infrastructure, this can lead to high carbon emissions. The provider we used, SURF, utilises 100% green power and does not contribute any $CO_2$ emission (van der Tak et al., 2021). Therefore, we provide a small insight into the estimated carbon emissions that result from the XAIFooler experiments by the original paper instead. Table 5 shows the runtime per document in the original study for XAIFooler and the computed total runtime per dataset and model. The maximum number of days for one experiment is 163.5. The total amount of attack runtime comes down to 478 days of computing time, excluding the four baselines, the training of the models and all other experiments.

Table 5: Average attack runtimes per document for XAIFooler as stated in the original paper and total attack runtime for the complete test sets.

| Model/ | | DistilBERT | | BERT | | RoBERTa | |
|---|---|---|---|---|---|---|---|
| Dataset | Test set size | Document (s) | Total (days) | Document (s) | Total (days) | Document (s) | Total (days) |
| IMDB | 25000 | 331.5 | 95.9 | 505.2 | 146.2 | 564.9 | 163.5 |
| S2D | 212 | 242.5 | 0.6 | 518.5 | 1.3 | 482.1 | 1.2 |
| GB | 6000 | 188.7 | 13.1 | 385.7 | 26.8 | 421.1 | 29.4 |

The carbon estimations are conducted using the MachineLearning Impact calculator presented in Lacoste et al. (2019). The hardware is RTX A4000 and 478 days are approximately 11466 hours. The provider is NVIDIA, but the carbon efficiency is not reported. Therefore, we estimate the carbon emissions for each provider based on regions that include south/east/central US (since Mississippi is in the southeast of the US). Although this estimate is not extremely precise, it does provide a range and can give the reader a rough idea of the environmental impact of this research. The results are presented in Table 5, showing that the carbon emissions roughly range from 450 to 1200 kg $CO_2$ equivalent. This is equal to driving an average ICE car 1810 to 4800 kilometres.

Table 6: Estimate of the carbon emissions for various providers and regions in the US for the complete test set (11466 h) as stated in the original paper.

| Provider | Google Cloud Platform | | Amazon Web Services | | | | Azure | | | CoreWeave | |
|---|---|---|---|---|---|---|---|---|---|---|---|
| Region of Compute (US) | Central | East | AWS GovCloud (Central) | AWS GovCloud (East) | East (N. Virginia) | East (Ohio) | Central | East | South Central | Central | East |
| Carbon Emitted (kg $CO_2$ eq) | 915.0 | 593.9 | 481.6 | 915.0 | 593.9 | 915.0 | 1187.9 | 593.9 | 738.4 | 674.2 | 449.5 |

