# OpenReview forum: "Reproducibility Study of “Are Your Explanations Reliable?” Investigating the Stability of LIME in Explaining Text Classifiers by Marrying XAI and Adversarial Attack"
_TMLR — Rejected by TMLR_

### Review · Reviewer_Ft4A · 2024-05-15

**Summary Of Contributions:**

This paper focuses on a single paper and tries to reproduce it. The topic is related to the explainable AI. They successfully reproduced and extended the results regarding the choice of RBO as similarity measure. And the third claim was partially reproducible due to constrained computational resources. However, they verify the third claim by observing similar trends on a small subset of the test data. In conclusion, all claims are supported in varying degrees through our reproducibility study.

**Audience:**

No

**Claims And Evidence:**

Yes

**Requested Changes:**

One significant change would be to add more relevant papers and reproduce them. If there is just one paper to reproduce, I tend to reject this paper.

**Strengths And Weaknesses:**

Strengths:

The overall topic is interesting, as XAI is very important nowadays.

Weaknesses:

Though TMLR does not require novelty, it still focuses on paper quality and significance. This paper has very limited impact to the field. The selected paper is just published, and showing that this paper's claims are solid does not contribute a lot to the field.

---

### Review · Reviewer_WaGy · 2024-07-29

**Summary Of Contributions:**

This work provides a detailed and critical reproduction study of the research paper titled "Are Your Explanations Reliable? Investigating the Stability of LIME in Explaining Text Classifiers by Marrying XAI and Adversarial Attack. This work serves as a meta-analysis on the early work by Burger et al. (2023). Although the study aims to validate and replicate the original findings, it exposes numerous challenges in the domain of reproducibility research itself.

The reproduction effort faced several significant challenges. A notable issue was the presence of numerous bugs, lack of clarity, absence of an established environment, and missing code segments in the original source code. Although the Git repository for the original paper was updated during the reproduction effort to fix bugs, not all issues were resolved. Afterall, this study supports the original paper's claims, particularly the inherent instability of LIME. However, the reproducibility of the experiments is limited due to unspecified hyperparameters in the original paper and the significant effort required to achieve reproducible results. The study concludes that while most of the claims are reproducible, doing so necessitates substantial understanding, effort, and computational resources.

To the end, this paper highlights the key aspects of reproducibility studies, including the importance of clear and complete documentation, responsive communication channels, and accessible codebases. It also sheds light on the environmental considerations linked to extensive computational research. Overall, this reproduction study serves as a valuable resource for understanding the challenges and best practices in achieving reproducible research in machine learning and data science.

**Audience:**

Yes

**Claims And Evidence:**

Yes

**Requested Changes:**

I hope authors could provide some meta-analysis on the "reproducibility issues" of this reproduction study. Especially, how can readers carried out a similar set of experiments using the same XAI algorithms on their own datasets. Could you please formulate the experimental protocols for a fair comparison? including data preparation, pre-processing, experiment metrics and setups.

**Strengths And Weaknesses:**

Pros:
- in-depth analysis of reproducbility issues of XAI research;
- communications with authors and bug-fixing to carry out experiments;
- well-written in a good structure.

Cons:
- My major concern is on the ``reproducibility issues of reproducibility studies''. I am wondering if this work is just yet-another-replicate of experiments or providing us a comprehensive ways to carry out our own experiments on the same set of algorithms.

---

### Review · Reviewer_AUGs · 2024-07-29

**Summary Of Contributions:**

The paper studies the reproducibility of the paper "Are your Explanations Reliable? Investigating the Stability of LIME in Explaining Text Classifiers by Marrying XAI and Adversarial Attack". The authors find that the original code does not work properly. When additionally worked by the authors, they could achieve to reproduce and even extend the results. In the end, all three claims the original paper made were supported by the author's study.

**Audience:**

No

**Broader Impact Concerns:**

No.

**Claims And Evidence:**

No

**Requested Changes:**

Mentioned in the Weaknesses. Even though I think reproducibility is an important work for the community, this paper does not meet the novelty for TMLR.

**Strengths And Weaknesses:**

Strengths:
- The authors find the lack of code implementation provided by the original authors.
- The authors reproduce the results and provide details of the implementation.

Weaknesses:
- Even though I think reproducibility is an important work for the community, this paper does not meet the novelty for TMLR. This paper mainly shows the details of the implementation that the original paper and code did not share, but not else. Only one new experiment was done to further support claim 2 in Section 4.4. I would recommend showing more new results and arguments that the original paper did not claim.
- For the study of Inherent Instability in Section 4.1, more evidence is needed to argue that the document size matters the most for the instability of LIME. By observing Figure 2, one could also hypothesize the reason for instability is the context of the dataset. Therefore, I would recommend experimenting with more datasets that have a similar size to GB/S2D/IMBD in order to support the argument.

---

### Review · Reviewer_QKfw · 2024-07-31

**Summary Of Contributions:**

- This paper conducted a comprehensive study by reproducing a prior work that investigated the stability issues of LIME. Throughout the experiments, the authors identified some key hyper-parameters that were unclear or downplayed in the original paper. This work confirms that most of the conclusions in the original paper are supported by their experiment results. The additional experiment results provide more information to better understand the original work.

**Audience:**

Yes

**Broader Impact Concerns:**

No concern

**Claims And Evidence:**

Yes

**Requested Changes:**

At the very least, the submission should address the two limitations mentioned above.

**Strengths And Weaknesses:**

**Strengths**

- A strength of this work is its detailed description of the experiment details and its effort to reproduce the results from prior work.
The additional experiment results can be viewed as another strength. As mentioned earlier, they provide more information for readers to better understand the original work.

**Weaknesses**

- First of all, the motivation for this work is unclear. For example, it is unclear whether the original work has an issue of reproducibility, for example, from the comments of follow-up work or general feedback from the community.
- Second, it is unclear whether the effort can benefit future research, especially since the experiments are mostly about re-confirming the conclusions in the original paper.

---

### Decision · Action_Editor_svgH · 2024-09-02

**Recommendation:** Reject

**Comment:**

This paper investigates the reproducibility of the published paper "Are your Explanations Reliable? Investigating the Stability of LIME in Explaining Text Classifiers by Marrying XAI and Adversarial Attack."  The authors found problems in the original codes, reproduced the results with additional efforts, and confirmed the original claims.

Although the authors found problems in terms of the reproducibility, which the authors of the original papers should have avoided, they are common problems in many papers that ML researchers have recognised. Therefore, what were reported in the paper is not of interest in the ML community.

in addition, the authors should have acknowledged the reviewer's service, and respond them.

**Audience:**

No

The paper reproduced results of a published paper, where the authors made efforts on debugging, parameter search, and using computational resources.  Although the authors found problems in terms of the reproducibility, which the authors of the original papers should have avoided, they are common problems in many papers that ML researchers have recognised. Therefore, what were reported in the paper is not of interest in the ML community.

**Claims And Evidence:**

Yes

The claims are supported in the experiment.

**Resubmission Of Major Revision:**

The authors may consider submitting a major revision at a later time.